# Partial Replacement of NaCl by KCl, MgCl_2_ and CaCl_2_ Chloride Salts in the Production of Sucuk: Effects on Volatile Compounds, Lipid Oxidation, Microbiological and Sensory Properties

**DOI:** 10.3390/foods12193525

**Published:** 2023-09-22

**Authors:** Derya Şimşek, Zeynep Feyza Yılmaz Oral, Rahimeh Jaberi, Mükerrem Kaya, Güzin Kaban

**Affiliations:** 1Department of Food Engineering, Faculty of Agriculture, Atatürk University, 25240 Erzurum, Türkiye; d.simsekcevik@gmail.com (D.Ş.); rahimeh.jaberi@gmail.com (R.J.); mkaya@atauni.edu.tr (M.K.); 2Department of Food Technology, Erzurum Vocational School, Atatürk University, 25240 Erzurum, Türkiye; zeynep.yilmaz@atauni.edu.tr; 3MK Consulting, Ata Teknokent, 25240 Erzurum, Türkiye

**Keywords:** fermented sausage, sucuk, salt, KCl, CaCl_2_, MgCl_2_, *Micrococcus/Staphylococcus*, volatile compound, TBARS

## Abstract

The effects of different chloride salt mixtures (I-control: 100% NaCl, II: 50:50—NaCl:KCl, III: 50:30:20—NaCl:KCl:MgCl_2_, IV: 50:30:20—NaCl:KCl:CaCl_2_, V: 50:30:10:10—NaCl:KCl:MgCl_2_:CaCl_2_) on the quality properties of sucuk (a dry fermented beef sausage) during ripening were investigated. Lactic acid bacteria reached 8 log cfu/g in the 3 days of fermentation in all treatments. However, salt mixtures including MgCl_2_ caused an increase in *Micrococcus/Staphylococcus*. The control group showed the lowest mean a_w_ value at the end of ripening. The salt mixture with 20% CaCl_2_ showed the lowest mean pH value of 4.97. The mean TBARS value varied between 6.34 and 6.97 µmol MDA/kg but was not affected by the salt mixtures (*p* > 0.05). According to the results of PCA, salt mixtures I, II and III had a positive correlation in PC1, and PC1 also separated salt mixtures with CaCl_2_ (IV and V) from other groups. In addition, a strong positive correlation between the control and III group (50:30:20—NaCl:KCl:MgCl_2_) for sensory properties was determined by heatmap clustering analysis. In addition, the principal component analysis showed that the control, II, and III groups had a stronger correlation with each other for volatile compounds.

## 1. Introduction

Sucuk is a traditional Turkish dry fermented sausage produced from beef and/or water buffalo meat, and it is widely consumed due to its typical sensory properties. Sucuk production is based on fermentation and drying/ripening. There is no heat treatment or smoking involved in the production of this product. While the initial fermentation temperature is between 12 and 26 °C, the processing time can be up to 20 days [1]. Sodium chloride (NaCl) is the major ingredient in sucuk production [2]. It plays an important role in sucuk and other fermented sausages in terms of technological and sensory qualities. NaCl contributes to the development of a desirable texture by increasing the solubility of myofibrillar proteins, and also gives a characteristic taste to the product. At the same time, NaCl limits the growth of undesirable microbiota by reducing initial water activity and favors the development of starter cultures. In addition, it has an important role in the aroma formation during ripening [3]. During the preparation of sucuk, 2–2.5% NaCl is usually added to the batter [2]. However, its amount in the final product can reach up to 5–6% due to dehydration occurring during ripening [4,5]. Values such as 6% in the end product were also determined for other types of fermented sausage [6]. For this reason, dry fermented sausages are a good source of sodium. In other words, NaCl is an important source of sodium in these meat products. At the same time, dry fermented sausages are reported to be among the sodium-rich food groups identified in the Global Sodium Benchmark for All Foods [7].

An excessive intake of sodium is the reason for the formation of many health problems [8] such as increase in hypertension, stroke, cerebrovascular and cardiovascular diseases [9]. Therefore, global health organizations recommend gradually reducing the use of salt in foods [10]. However, a simple salt reduction can adversely affect the microbial safety and quality of fermented sausages, because NaCl reduces water activity, and ensures microbiological stability and a salty taste. Therefore, partial replacement of NaCl with potassium chloride (KCl), magnesium chloride (MgCl_2_) and calcium chloride (CaCl_2_) has been the main approach to reduce sodium content [11]. Different chloride salts have different impacts in fermented sausages. This is due to the ionic strength and exposed charged groups of the neutral salts [12]. For example, using high concentrations of KCl reduces the salty taste and promotes a metallic and bitter taste in the products [13], while CaCl_2_ and MgCl_2_ cause an undesirable odor formation [14].

Several studies were carried out to determine the impacts of different chloride salt mixtures on microbiological, physicochemical and sensorial properties of dry and semi-dry fermented sausages such as Harbin dry sausage, Italian salami, Brazilian dry fermented sausage and heat-treated sucuk [13,15,16,17,18,19,20,21]. However, there is no study on the reduction of sodium content in the production of sucuk, a dry fermented sausage. At the same time, as planned in this study, there is no study on the substitution of other chloride salts with each other in a fermented sausage during ripening, provided that the NaCl ratio in salt mixture is kept at 50%. In this study, sucuk batter was prepared with five different salt mixtures (I-control: 100% NaCl, II: 50:50—NaCl:KCl, III: 50:30:20—NaCl:KCl:MgCl_2_, IV: 50:30:20—NaCl:KCl:CaCl_2_, V: 50:30:10:10—NaCl:KCl:MgCl_2_:CaCl_2_). Microbiological and physicochemical analyses were performed for each group on certain days of the ripening process (day 0, 1, 3, 5, 7, 9 and 12). In addition, the final products were analyzed in terms of volatile compounds and sensory properties.

## 2. Materials and Methods

### 2.1. Material

Beef meat, beef fat and sheep tail fat were taken from the meat slaughterhouse of the Meat and Dairy Institution at Erzurum. Raw materials were cut into small pieces after trimming and stored at −20 °C until production. As starter cultures, *Lactiplantibacillus plantarum* GM77 and *Staphylococcus xylosus* GM92 [22] were used, and they were grown in de Man Rogosa Sharpe (Merck, Darmstadt, Germany) broth and in tryptic soy broth (Merck), respectively, for 24 h at 30 °C. They were added in batters at approximately 10^7^ cfu/g and 10^6^ cfu/g, respectively.

### 2.2. Sucuk Production

In sucuk production, the formulation consisted of the meat–fat mixture (80% beef meat, 10% sheep tail fat and 10% beef fat), 10 g garlic, 4 g sucrose, 2.5 g pimento, 5 g black pepper, 7 g red pepper, 9 g cumin and 150 mg/kg sodium nitrite. Two independent batters of sausages were prepared, and five sucuk treatments were performed in each batter: I (control, 100% NaCl), II (50:50—NaCl:KCl), III (50:30:20—NaCl:KCl:MgCl_2_), IV (50:30:20—NaCl:KCl:CaCl_2_) and V (50:30:10:10—NaCl:KCl:MgCl_2_:CaCl_2_). From each salt mixture, 25 g salt per kg was used in the production. A laboratory-type cutter (MADO Typ MTK 662, Dornhan, Schwarzwald) was used to prepare the sucuk batter, and a laboratory-type piston filler (MADO Typ MTK 591) was used for filling the batter. After filling, the samples were taken to a climate chamber (Reich, Thermoprozestechnik GmbH, Schechingen, Germany). The ripening program was as follows: for 3 days at 22 ± 1 °C and 90 ± 2% relative humidity (RH), for the 4th and 5th day at 20 °C ± 1 °C and RH 85 ± 2%, and for 6 days at 18 °C ± 1 °C and 80 ± 2%, respectively. 

### 2.3. Microbiological Analysis

Samples of 25 g were added to 225 mL physiological saline (0.85 NaCl%) and homogenized in a stomacher (Lab Stomacher, London, UK) for 1 min. In determining the number of lactic acid bacteria, de Man Rogosa Sharpe Agar (Merck, Darmstadt, Germany) was used and incubation was carried out at 30 °C for 48 h under anaerobic conditions (Anaerocult A, Merck, Darmstadt, Germany). Mannitol Salt Phenol Red Agar (Merck) was used for the enumeration of *Micrococcus/Staphylococcus*, and the plates were incubated aerobically at 30 °C for 48 h. Enterobacteriaceae were determined on Violet Red Bile Dextrose Agar (Merck), and the plates were incubated at 30 °C for 48 h under anaerobic conditions (Anaerocult A, Merck) [23].

### 2.4. a_w_, pH and TBARS 

A water activity (a_w_) device (Novasina, TH-500 a_w_ Sprint, Pfaffikon, Switzerland) was used to determine the a_w_ values of the samples at 25 °C. For pH measurement, 10 g of sample was weighted and 100 mL of distilled water was added to it. Then, it was homogenized with ultra-turrax, and the pH values of the samples were determined by pH-meter (ATI ORION 420, Orion Inc., Boston, MA, USA) [24].

For the determination of the TBARS value, 2 g of homogenized sample was mixed with 12 mL TCA solution with EDTA and propyl gallate, and homogenized. After filtration with Whatman 1 filter, 0.02 M thiobarbituric acid solution was added to 3 mL of fitrate, and kept in a boiling water bath for 40 min. The mixture was then centrifuged at 2000× *g* for 5 min. The absorbance was then measured at 530 nm. The TBARS value was expressed as μmol MDA/kg sample [25].

### 2.5. Sensory Evaluation 

The sensory properties of the final products were evaluated by 10 semi-trained panelists who are academic staff. The panelists were given information on sampling, analyzing and interpreting stimuli, and providing responses as respective scores. The panelists were also informed in detail about the score card (9: very good, 1: very bad) and the evaluation procedures. The color, texture, odor, taste and overall acceptability were evaluated. The samples were sliced approximately 1 cm thick, and a portion of each sample was randomly selected and served to the panelists at room temperature. After each sample evaluation, the panelists were provided with water and bread. Each sample was evaluated individually in two independent replicates.

### 2.6. Volatile Compounds

The homogenized sample was weighed as 5 g into a 40 mL vial (Supelco, Bellefonte, PA, USA). The vial was placed into a thermal block (Supelco) for 1 h at 30 °C to collect the volatile compounds in the headspace. For the adsorption of compounds, carboxen/polydimethylsiloxane (75 µm, Supelco) fiber was placed in the vial and kept for 2 h. Gas chromatography/mass spectrometry (Agilent 6890N/Agilent 5973, Santa Clara, CA, USA) was used to identify volatile compounds. The carrier gas was helium and DB-624 (J&W Scientific, 60 m × 0.25 mm × 1.4 μm) was used as column. The oven temperature of GC was initially set at 40 °C for 6 min, and then gradually increased to 210 °C, and kept for 12 min. The injector port was in splitless mode, and the interface was kept at 280 °C. The library of mass spectrometry (NIST, WILEY and FLAVOR) and standard substances were used for the identification of compounds, and the Kovats index was determined using the standard mix (Paraffins mix, Supelco 44585-U, Bellefonte, PA, USA). The results were given as Au × 10^6^ [1].

### 2.7. Statistical Analysis 

A completely randomized design with two replicates was used for the experiments. Two independent batters of sausages (replicates) were prepared on different days, and a total of five treatments of sausages were prepared in each batter. All measurements were performed twice. An analysis of variance was performed using the SPSS vs. 20 (IBM SPSS, Chicago, IL, USA) to analyze the effects of factors and the interaction between factors (salt mixture and ripening time). Duncan’s multiple range test at *p* < 0.05 was used to compare the means of significant sources of variation. Sensory property and volatile compound analyses were carried out in the final products. In addition, a principal component analysis (PCA) was performed using the Unscrambler software (CAMO vs. 10.1, Oslo, Norway) to determine the relationship between the salt mixture and volatile compounds. The differential profile (cluster heat map) of the sensory analysis in the salt mixture groups was analyzed using heat mapper (http://www.heatmapper.ca) (accessed on 1 September 2023).

## 3. Results and Discussion

### 3.1. Microbiological Properties

The ripening time and the interaction of salt mixture and ripening time had a very significant effect on lactic acid bacteria (LAB) (*p* < 0.01) (Table 1). The growth of LAB was slower in the first 24 h of fermentation in the control group and IV group compared to the other groups. However, on the 3rd day of ripening, the LAB counts showed very close values to each other and similar results were obtained on other days of ripening for all treatments (Figure 1). In sucuk processing, the early stage of fermentation is very important for the growth of lactic acid bacteria. In this study, lactic acid bacteria reached 8 log cfu/g in the 3 days of fermentation in all treatments (Figure 1). Accordingly, different chloride salts have no influence on the development of LAB, which is technologically important in sausages. LAB contribute to safety and sensory properties through the production of lactic acid [3]. Likewise, Gimeno et al. [20] also reported that changing the NaCl ratio in fermented sausages does not have a significant effect on lactic acid bacteria.

The salt mixture, ripening time and their interaction had a very significant effect on *Micrococcus/Staphylococcus* (*p* < 0.01) (Table 1). In the first 24 h of fermentation, lower numbers were detected in the control and III groups compared to the other salt mixtures (Figure 2). Similarly, Dos Santos et al. [15] also determined that a 50% replacement of NaCl with CaCl_2_ caused a decrease in the number of *Micrococcoceae* in fermented sausages. As can be seen from Figure 2, the salt mixtures containing 20% MgCl_2_ (group III) and 10% MgCl_2_ (group V) generally gave a higher count of *Micrococcus/Staphylococcus*, albeit slightly, during the ripening period. These results showed that the use of MgCl_2_ promotes the growth of *Micrococcus/Staphylococcus* in sucuk during ripening (Figure 2). Moreover, Campagnol et al. [6,11] reported that different chloride salts used with salt did not have a significant effect on Micrococcaceae.

Enterobacteriaceae remained under <2 log cfu/g in all treatments during ripening. Similar results were reported in previously studies on sucuk [1,26].

### 3.2. a_w_, pH and TBARS

The salt mixture had a significant effect on the a_w_ value of sucuk. Although the lowest mean a_w_ value was found in the group with 100% NaCl, the differences between the groups were very small (Table 1). Similarly, Teixeria et al. [27] reported that salt replacement had a very low effect on the a_w_ value of Bísaro pork sausage. 

The interaction of factors had a significant effect on the a_w_ value of sucuk (Table 1). In this study, the a_w_ value decreased below 0.90 on the 12th day in all groups. This decrease in the a_w_ value contributes significantly to the stability and safety of sucuk. This result showed that the salt mixtures did not cause a significant difference in drying even though they partially differed compared to the control (Figure 3). The control group showed lower a_w_ values than the other groups during ripening. This situation is thought to be related to the penetration of chloride salts. In fact, Dos Santos et al. [28] stated that the 50:50—NaCl:CaCl_2_ mixture gave a higher a_w_ value than the control during the ripening of fermented sausage, and this result was due to the strong bonding between CaCl_2_ and meat proteins. Similarly, Gimeno et al. [29] reported that the a_w_ value was higher in a dry fermented sausage type with reduced sodium content compared to the control group (100% NaCl). On the other hand, Campagnol et al. [6] stated that no difference was observed between the group containing 100% NaCl and group II in terms of a_w_ value. It has also been reported that the use of 50% KCl did not affect the water activity value and there was no problem in the drying of the product [15]. On the other hand, in a study conducted on semi-dry fermented sausage (heat-treated sucuk), it was reported that groups with high KCl levels (>50%) showed higher a_w_ values than the other groups (100:0, 75:25 and 50:50—NaCl:KCl), and it was indicated that this may be explained by potassium reacting with muscle surface proteins to prevent sodium penetration [18]. 

Individually, the salt mixture and ripening time had a very significant effect on the pH value (*p* < 0.01), whereas the interaction of salt mixture and ripening time had no significant effect (*p* > 0.05) on the pH value of sucuk (Table 1). The mean pH value in group IV was found to be lower than in the other groups (Table 1). According to these results, pH showed a significant decrease in the presence of 20% CaCl_2_. Similarly, Dos Santos et al. [15] found that the use of CaCl_2_ caused a decrease in the pH value. Gimeno et al. [29] stated that the salt mixture consisting of 1% NaCl + 0.55% KCl + 0.23% MgCl_2_ + 0.46% CaCl_2_ caused more acidification than the control group, and this significantly contributed to product stability. The rate and extent of lactic acid production by fermentation are very important for the growth and survival of the sausage microbiota. In this study, the *Micrococcus/Staphylococcus* count decreased in the groups containing CaCl_2_ due to these microorganisms’ sensitivity to a low pH value (Figure 2). On the other hand, the group II gave higher mean pH value than the other groups (Table 1). Similarly, Guardia et al. [17] indicated that there was a statistically significant increase in the pH value compared to the control group as a result of the 50% replacement of the sodium ratio with KCl. However, Campagnol et al. [6,11] reported that no significant change in pH was observed with the use of KCl.

Due to the good growth of the starter culture, the mean pH dropped below 5.0 on the 3rd day. After the 7th day of the ripening period, an increase in pH was observed (Table 1) due to the basic character compounds which formed as a result of proteolysis [1]. 

The salt mixture had no significant effect on the TBARS value of sucuk, while the ripening time had a very significant effect on this value (*p* < 0.01) (Table 1). As can be seen from Table 1, partial increases were observed depending on the progress of the ripening and the highest mean TBARS value was detected on the 12th day in all groups. In the present study, the salt mixture had no effect on the TBARS value of sucuk. A similar result was observed in Bísaro pork sausage (a type of dry fermented sausage) [27]. Wen et al. [19] also found that there was no difference in the TBARS value between the control (100% NaCl) and the 70% NaCl + 30% KCl group at the end of ripening process. On the other hand, the effects of chloride salts on TBARS can vary during the ripening and storage of fermented sausages. In fact, Dos Santos et al. [30] reported that there were differences between the control (100% NaCl) and different salt mixtures (50:50—NaCl:CaCl_2_, 50:25:25—NaCl:KCl:CaCl_2_) at the end of the production of salami in terms of TBARS, but there were no differences between the groups at the end of the storage.

### 3.3. Sensory Evaluation

In order to investigate the general differences between the groups, a cluster analysis based on the sensory characteristics of the groups was performed. To show the similarities between the groups, a dendrogram was made using a heat map clustering algorithm. High, medium and low expression levels were represented by the heatmap colors. To express the heat map, the centroid linkage for clustering and Pearson correlation for distance measurement were used. Figure 4 shows the two main clusters. The first cluster included the I (control) and III (50% NaCl/30% KCl/20% MgCl_2_) groups. The second cluster contained two sub-groups, of which the first one included the V group, and the second one the II and IV groups. According to the results, the I and III groups had more correlation with each other for sensory properties. On the other hand, there were two clusters for sensory properties, and texture was separated from the other properties. The second cluster included color, odor, taste and general acceptability. General acceptability closely correlated with taste and color. Indeed, it was emphasized that the use of 50% KCl significantly reduced the taste, aroma and general acceptability scores [11]. In a study conducted on heat-treated sucuk, which differs from sucuk due to its short fermentation period and different physicochemical properties, it was reported that the use of a KCl ratio of more than 50% instead of NaCl in heat-treated sucuk (a kind of semi-dry fermented sausage) was not possible, since the sensory properties, in particular the taste, are adversely affected [18]. On the other hand, it was reported that the use of 50% CaCl_2_ significantly reduced the aroma scores, and this result was due to CaCl_2_ increasing lipid oxidation [15]. In this study, it was determined that the control group (100% NaCl) was more correlated with group III in terms of sensory properties, and also general acceptability was more correlated with taste and odor. 

### 3.4. Volatile Compounds 

In sucuk groups, aldehydes, ketones, acids, alcohols, sulphur compounds, aromatic hydrocarbons, aliphatic hydrocarbons and terpenes have been identified (Table 2). The using rate of NaCl is effective on the activity of many muscles and microbial enzymes which contribute to flavor development in fermented sausages [15]. The different chloride salts had a very significant (*p* < 0.01) effect on acetaldehyde, 2-propene-1-thiol, 3,3-thiobis-1-propene and propanoic acid, and this factor had a significant (*p* < 0.05) effect on 2,3-butanedione, acetic acid, allyl methyl sulfide, 2-butanone-3-hydroxy, hexanal, p-xylene, butyl propionate, dodecane, 1-decanol, and tridecane. Dos Santos et al. [15] stated that the substitution of NaCl with KCl and decrease in the sodium chloride ratio had no significant effect on volatile compounds in a Brazilian-type dry fermented sausage, and they indicated also that the use of CaCl_2_ increased the formation of hexanal and 2-heptanal resulting from lipid oxidation during both processing and storage.

In this study, the group with 20% MgCl_2_ showed the highest abundance for acetaldehyde and this value was not statistically different from the control group (Table 2). The lowest abundance of acetaldehyde was determined in group V. Hexanal, which causes rancid odors in foods and is a dominant breakdown product of lipid oxidation of n-6 fatty acids [31], showed the highest abundance in the control. The difference between the other groups was not statistically significant. It is thought that NaCl gives a higher hexanal value because it is a prooxidant. It was reported that NaCl is an important prooxidant in meat and meat products [32]. 

Acetic acid and propanoic acid were determined as acids in sucuk samples. The lowest abundance of acetic acid was determined in the control group and the other groups were not statistically different from each other (*p* > 0.05). Dos Santos et al. [15] also determined that the 50:50—NaCl:KCl salt mixture increased the amount of acetic acid and reported that this compound had a significant effect on the aroma of the product. Similarly, Qin et al. [33] indicated that KCl substitution increased the levels of acids. Acetic acid could be produced by homofermentative lactic acid bacteria and staphylococci. In addition, this acid could be formed as a result of fatty acid oxidation and alanine catabolism [34]. Another acid determined in sucuk samples was propanoic acid, and its abundance showed a significant decrease in the IV and V groups. As can be seen from Table 2, group V showed the lowest abundance of propanoic acid. 

The use of salt mixture had a significant effect on 1-decanol (Table 2). Group II with 50% KCl treatment gave the highest abundance of 1-decanol. However, this value was not statistically different from that of the control group (*p* > 0.05). However, this compound could not be detected in the CaCl_2_ treatment. The use of different chloride salts had a significant effect on butyl propionate (*p* < 0.05). The lowest abundance was determined in the CaCl_2_ treatment. However, this value was not statistically different from the value of group II (50: 50—NaCl:KCl). Esters in meat products are generally formed as a result of esterification of carboxylic acids and alcohols. Many esters have also been detected in studies to determine the effects of different factors on the volatile compounds of sucuk [1,35]. 

As can be seen from Table 2, group II with 50% KCl treatment gave the highest abundance of 2,3-butanedione, and this value showed a statistical difference from other groups. Also, the highest abundance of 3-hydroxy-2-butanone (acetoin) was detected in group III with 20% MgCl_2_, and this value did not differ from the control group and group IV (20% CaCl_2_) (*p* > 0.05) (Table 2). 

Allyl methyl sulfide and 3,3-thiobis-1-propene were affected by the use of salt mixtures at *p* < 0.05 and *p* < 0.01, respectively. The highest abundance for both compounds was determined in the control, and the lowest abundance was determined in group V (Table 2). Various sulphur compounds have also been detected in studies on sucuk and other fermented sausages [1,34,35,36]. It is stated that sulfide compounds may be formed as a result of amino acid catabolism [36], as well as garlic [6], or a reaction between garlic and other components [34].

Among aliphatic hydrocarbons, dodecane, tridecane and tetradecane were affected by the salt mixture (*p* < 0.05). The highest abundance for these compounds was found in the control group. The differences between the other groups were not statistically significant (*p* > 0.05) (Table 2). Aliphatic hydrocarbons have very low effects on flavor due to their high threshold values. Various aliphatic hydrocarbons have also been detected in other studies on sucuk. However, their amounts are generally low, as in this study [1,35,37]. Another chemical group determined in samples was aromatic hydrocarbons. The source of aromatic hydrocarbons also varies considerably. For example, it is stated that toluene may be formed as a result of lipid degradation, or may originate from grasses used as animal feed [38], or may originate from amino acid catabolism [36]. In the present study, only p-xylene was affected by the use of different chloride salts (*p* < 0.05). The highest abundance was determined in the control group. However, there was no difference between the control (group I) and group III with 20% MgCl_2_ (Table 2).

Terpenes were not affected by the replacement of NaCl (*p* > 0.05) (Table 2). Red pepper, black pepper, cumin, pimento and garlic used in sucuk production constitute an important source of volatile compounds. Terpenes constitute a very important part of the volatile compounds originating from spices. Many terpene compounds have been identified in studies on sucuk [1,35,37].

The result of the PC analysis of relationship between the groups with different salt mixtures and volatile compounds is given in Figure 5. The PC1 explained 84% of the total variation, and 95% by the first two components (Figure 5). The groups containing 100% NaCl (control), 50:50—NaCl:KCl (group II) and 50:30:20—NaCl:KCl:MgCl_2_ (group III) were on the positive side of PC1. Groups IV and V with CaCl_2_ were located on the negative side of PC1. In addition, most volatile compounds were located on the positive side of PC1, and negative correlations were observed with the groups with CaCl_2_. According to these results, NaCl and the combinations NaCl:KCl and NaCl:KCl:MgCl_2_ showed similar characteristics in terms of volatile compounds. Groups IV and V with CaCl_2_ were also separated from these groups by PC1. A similar result was also observed in the heat map analysis for sensory properties.

## 4. Conclusions

The results indicated that a replacement of 50% of NaCl with other chloride salts did not have an effect on lactic acid bacteria. The salt mixture containing 50:30:20—NaCl:KCl:MgCl_2_ resulted in slightly higher *Micrococcus/Staphylococcus* numbers, which did not adversely affect the product properties. In addition, the mixtures with CaCl_2_ caused a decrease in the pH value. The a_w_ was found to be under 0.90 at the end of the ripening period in all groups. TBARS were not affected by the NaCl replacement. Based on these results, it can be concluded that the microbiological and physicochemical characteristics of sucuk were not adversely affected by the salt mixtures selected. Considering the volatile compounds and sensory evaluation results, it can be concluded that replacements of 50% NaCl with 50% KCl and with 30% KCl + 20% MgCl_2_ are good alternatives to reduce the sodium content in sucuk.

## Figures and Tables

**Figure 1 foods-12-03525-f001:**
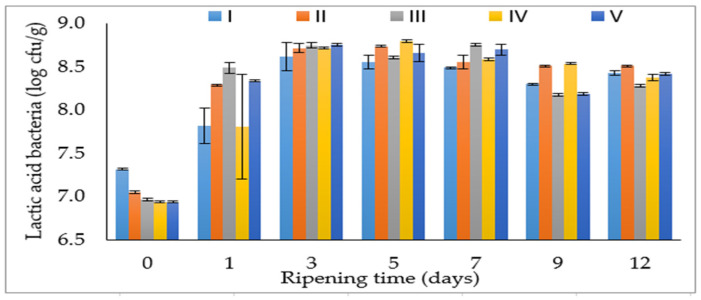
The effect of the interaction of salt mixture × ripening time on LAB counts of sucuk (I: 100% NaCl-Control, II: 50:50—NaCl:KCl, III: 50:30:20—NaCl:KCl:MgCl_2_, IV: 50:30:20—NaCl:KCl:CaCl_2_, V: 50:30:10:10—NaCl:KCl:MgCl_2_:CaCl_2_).

**Figure 2 foods-12-03525-f002:**
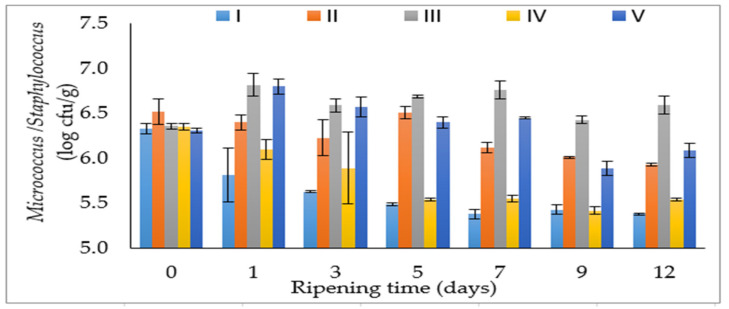
The effect of the interaction of salt mixture × ripening time on *Micrococcus/Staphylococcus* counts of sucuk (I: 100% NaCl-Control, II: 50:50—NaCl:KCl, III: 50:30:20—NaCl:KCl:MgCl_2_, IV: 50:30:20—NaCl:KCl:CaCl_2_, V: 50:30:10:10—NaCl:KCl:MgCl_2_:CaCl_2_).

**Figure 3 foods-12-03525-f003:**
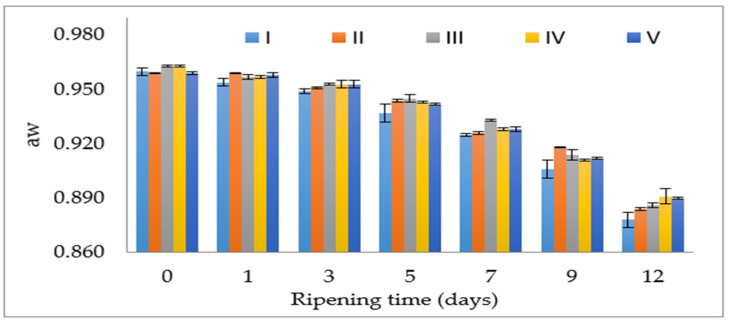
The effect of the interaction of salt mixture × ripening time on the a_w_ value of sucuk (I: 100% NaCl-Control, II: 50:50—NaCl:KCl, III: 50:30:20—NaCl:KCl:MgCl_2_, IV: 50:30:20—NaCl:KCl:CaCl_2_, V: 50:30:10:10—NaCl:KCl:MgCl_2_:CaCl_2_).

**Figure 4 foods-12-03525-f004:**
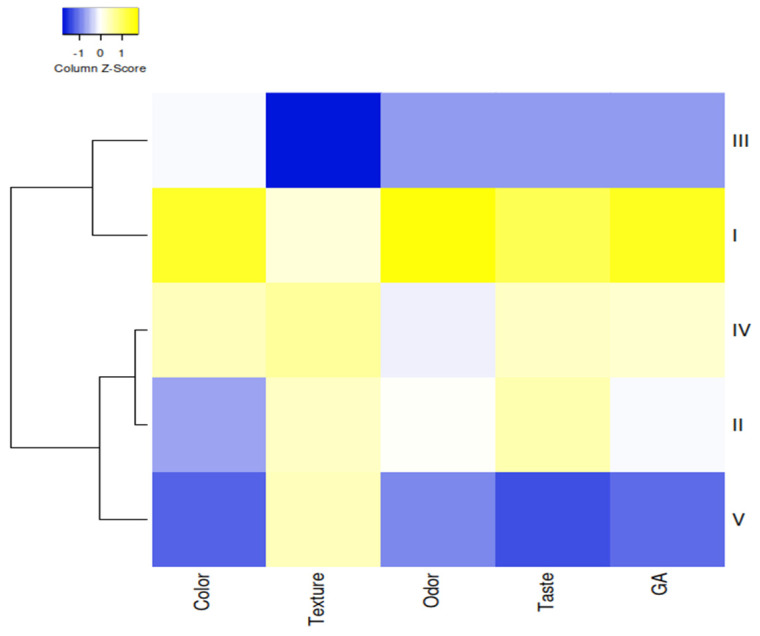
Heat map of Pearson rank correlation between groups containing different chloride salts for sensorial properties (I: 100% NaCl-Control, II: 50:50—NaCl:KCl, III: 50:30:20—NaCl:KCl:MgCl_2_, IV: 50:30:20—NaCl:KCl:CaCl_2_, V: 50:30:10:10—NaCl:KCl:MgCl_2_:CaCl_2_, GA: general acceptability).

**Figure 5 foods-12-03525-f005:**
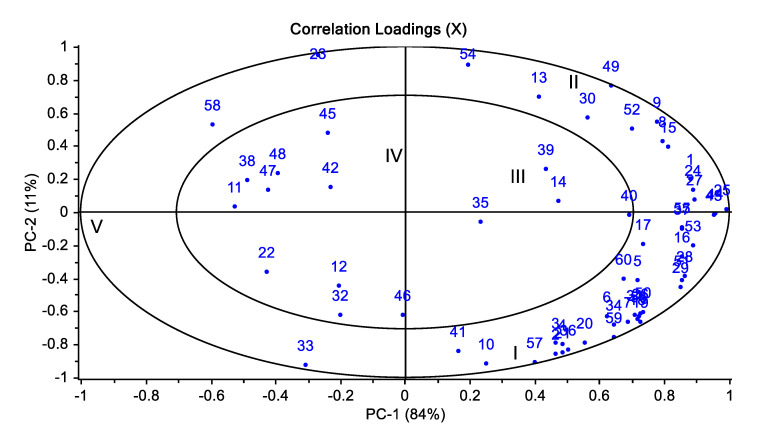
The changes in salt mixture and ripening time on volatile compounds of sucuk (the numbering is given in Table 1) (I: 100% NaCl-Control, II: 50:50—NaCl:KCl, III: 50:30:20—NaCl:KCl:MgCl_2_, IV: 50:30:20—NaCl:KCl:CaCl_2_, V: 50:30:10:10—NaCl:KCl:MgCl_2_:CaCl_2_).

**Table 1 foods-12-03525-t001:** Overall effects of salt mixture and ripening time on lactic acid bacteria (LAB), *Micrococcus/Staphylococcus* (M/S), a_w_, pH and TBARS in sucuk (mean ± SEM).

	LAB (log cfu/g)	M/S (log cfu/g)	a_w_	pH	TBARS (µmol MDA/kg)
Salt Mixture (SM)					
I	8.22	5.64 e	0.930 b	5.08 b	6.62
II	8.34	6.25 c	0.934 a	5.14 a	6.66
III	8.29	6.61 a	0.936 a	5.04 bc	6.52
IV	8.25	5.77 d	0.935 a	4.97 d	6.97
V	8.29	6.36 b	0.934 a	5.02 c	6.34
Significance	ns	**	*	**	ns
Ripening Time (RT)					
0	7.04 e	6.37 a	0.960 a	5.79 a	6.60 cb
1	8.15 d	6.39 a	0.957 b	5.47 b	6.13 c
3	8.71 a	6.18 b	0.952 c	4.76 e	6.03 c
5	8.47 b	6.12 bc	0.942 d	4.80 cd	6.88 b
7	8.62 a	6.05 c	0.928 e	4.80 cd	6.94 ab
9	8.34 c	5.83 d	0.912 f	4.83 d	6.31 c
12	8.40 bc	5.91 d	0.886 g	4.88 c	7.47 a
Significance	**	**	**	**	**
**SM × RT**	**	**	**	ns	ns
SEM	0.01	0.01	0.001	0.01	0.07

^a–g^ Any two means in the same column having the same letters in the same section are not significantly different at *p* > 0.05, **: *p* < 0.01, *: *p* < 0.05, ns: not significant, SEM: standard error of the mean. (I: 100% NaCl-Control, II: 50:50—NaCl:KCl, III: 50:30:20—NaCl:KCl:MgCl_2_, IV: 50:30:20—NaCl:KCl:CaCl_2_, V: 50:30:10:10—NaCl:KCl:MgCl_2_:CaCl_2_).

**Table 2 foods-12-03525-t002:** Volatile compounds of sucuk produced using different chloride salts (mean ± SEM) (Au × 10^6^).

Compounds	No	KI	Salt Mixtures					SEM	Sig.
			I	II	III	IV	V		
**Aliphatic hydrocarbons**									
Undecane	1	1100	0.97	1.24	1.06	0.59	0.35	0.12	ns
Dodecane	2	1200	3.70 a	0.57 b	0.63 b	0.34 b	0.33 b	0.27	*
Tridecane	3	1300	4.59 a	1.64 b	0.83 b	0.81 b	0.93 b	0.24	*
Tetradecane	4	1400	3.27 a	1.30 b	1.11 b	0.64 b	0.90 b	0.15	*
Pentadecane	5	1500	0.61	0.47	0.49	0.21	0.26	0.08	ns
**Aromatic hydrocarbons**									
Toluene	6	796	2.28	1.72	1.70	1.22	1.37	0.14	ns
p-xylene	7	898	1.36 a	0.24 c	0.91 ab	0.40 cb	0.00 c	0.09	*
Styrene	8	935	0.47	0.60	0.71	0.63	0.21	0.10	ns
1-methyl-2-(1-methylethyl)-benzene	9	1062	65.34	81.45	67.54	68.58	53.0	12.22	ns
1-methoxy-4-(1-propenyl)-benzene	10	1230	1.69	0.54	0.59	0.88	0.75	0.25	ns
1-methoxy-4-(2-propenyl)-benzene	11	1238	0.50	0.41	0.92	0.66	0.84	0.25	ns
1,2-dimethoxy-4-(2-propenyl)-benzene	12	1482	1.81	1.58	1.76	1.91	1.77	0.15	ns
**Ketones**									
2,3-Butanedione	13	657	1.15 b	4.38 a	1.33 b	2.05 b	0.94 b	0.22	*
3-hydroxy-2-butanone	14	779	6.44 ab	5.22 b	8.63 a	8.13 a	4.57 b	0.29	*
6-methyl-5-hepten-2-one	15	1050	0.37	0.44	0.52	0.44	0.21	0.09	ns
**Aldehydes**									
Acetaldehyde	16	623	12.49 a	9.00 b	12.73 a	9.97 a	5.45 c	0.21	**
Pentanal	17	742	1.32	1.15	0.81	1.22	0.62	0.34	ns
Hexanal	18	849	4.54 a	2.25 b	1.92 b	2.03 b	0.85 b	0.24	*
Heptanal	19	955	1.22	0.59	0.63	0.65	0.29	0.09	ns
Octanal	20	1051	1.94	0.75	0.86	0.42	0.49	0.17	ns
Nonanal	21	1143	5.14	1.05	1.16	0.83	0.64	0.55	ns
2-methyl-3-phenylpropanal	22	1333	4.62	3.45	5.03	5.14	5.00	0.82	ns
**Acids**									
Acetic acid	23	710	35.08 b	46.83 a	44.46 a	46.65 a	44.30 a	0.70	*
Propionic acid	24	889	7.61 a	8.86 a	8.60 a	4.58 b	2.93 c	0.17	**
**Sulphur compounds**									
2-propene-1-thiol	25	570	93.09 a	89.70 a	92.59 a	76.47 b	37.12 c	1.52	**
Allyl methyl sulfide	26	730	12.02 a	8.17 b	7.09 b	7.64 b	5.35 b	0.46	*
3-*methylthiophene*	27	874	1.38	1.40	1.17	1.37	0.80	0.07	ns
3,3-thiobis -1-propene	28	888	11.20 a	8.27 b	8.72 ab	4.78 c	3.33 c	0.34	**
Disulfide methyl 2-propenyl	29	958	2.94	2.23	2.33	1.66	1.32	0.13	ns
Diallyl disulfide	30	1138	8.08	9.02	8.64	7.89	7.90	0.66	ns
**Esters**									
Ethyl acetate	31	648	3.18	0.92	2.08	0.29	0.00	0.34	ns
Butyl propionate	32	952	1.33 a	0.67 b	1.30 a	0.38 b	1.33 a	0.07	*
3-hexenoic acid. methyl ester	33	954	0.76	0.59	0.68	0.66	0.73	0.07	ns
Propyl hexanoate	34	1151	4.62	1.96	3.44	1.58	1.37	0.29	ns
2.4-hexadienoic acid ethyl ester	35	1142	2.05	1.71	2.66	2.18	1.89	0.35	ns
Hexyl *butanoate*	36	1221	4.62	1.12	1.04	0.73	0.58	0.41	ns
Hexyl hexanoate	37	1133	1.05	1.08	0.76	0.43	0.29	0.15	ns
**Alcohols**									
*Ethyl alcohol*	38	539	0.35	0.47	0.64	0.29	0.66	0.07	ns
2-ethyl-1-*hexanol*	39	1084	0.21	0.19	0.51	0.39	0.11	0.08	ns
1-decanol	40	1218	0.68 ab	0.91 a	0.24 bc	0.00 c	0.00 c	0.07	*
4-(1-*methylethyl*)-benzenemethanol	41	1383	3.64	2.56	3.29	3.18	3.03	0.41	ns
**Terpenes**									
α-thujene	42	944	0.90	0.81	1.31	1.23	1.08	0.16	ns
α-pinene	43	950	3.33	3.13	3.62	3.21	2.16	0.22	ns
Camphene	44	970	0.58	0.59	0.43	0.42	0.15	0.04	ns
β-thujene	45	994	0.77	0.90	1.01	1.35	0.98	0.19	ns
β-pinene	46	996	8.41	7.43	7.95	8.50	7.95	0.57	ns
β-myrcene	47	1005	8.50	7.79	11.51	11.45	10.65	1.57	ns
α-*phellandrene*	48	1019	4.16	3.98	7.04	6.78	6.10	1.15	ns
3-carene	49	1026	10.92	17.19	14.52	14.55	9.22	2.27	ns
α-terpinene	50	1030	3.67	2.33	1.91	1.80	1.21	0.45	ns
D-limonene	51	1054	47.37	37.95	32.62	34.53	22.75	6.88	ns
β-phellandrene	52	1065	3.17	3.96	3.30	3.02	2.74	0.61	ns
ɣ-terpinene	53	1105	34.69	31.84	25.70	28.40	18.79	6.41	ns
α-terpinolene	54	1131	0.01	0.52	0.18	0.25	0.14	0.10	ns
p-cymenene	55	1157	3.23	3.27	2.98	2.56	2.44	0.57	ns
Linalool	56	1161	11.73	8.55	8.38	8.88	6.98	1.95	ns
Terpinen-4-ol	57	1233	3.23	0.91	1.18	1.07	1.03	0.54	ns
α-terpineol	58	1258	0.83	0.99	1.07	1.27	1.17	0.25	ns
Copaene	59	1433	0.80	0.45	0.56	0.48	0.38	0.17	ns
Caryophyllene	60	1490	5.40	4.44	5.06	5.26	4.09	0.61	ns

^a–c:^ Different letters in a same row mean that the values differ significantly at (*p* < 0.05), *: *p* < 0.05, **: *p* < 0.01, ns: not significant, SEM: standard error of the mean. KI: Kovats index calculated for DB-624 capillary column installed on a GC/MS. (I: 100% NaCl-Control, II: 50:50—NaCl:KCl, III: 50:30:20—NaCl:KCl:MgCl_2_, IV: 50:30:20—NaCl:KCl:CaCl_2_, V: 50:30:10:10—NaCl:KCl:MgCl_2_:CaCl_2_).

## Data Availability

The data presented in this study are available in the article.

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
