# Peer review of "Partial Replacement of NaCl by KCl, MgCl2 and CaCl2 Chloride Salts in the Production of Sucuk: Effects on Volatile Compounds, Lipid Oxidation, Microbiological and Sensory Properties"

_foods, 2023, doi:10.3390/foods12193525_

Round 1

Reviewer 1 Report

Article may be accepted after addressing the following comments:

1. Add some numerical data in the abstract portion.

2. Add some keywords related to the properties studied.

3. Rewrite the introduction portion, start with Sucuk, then discuss the role of different salts on the sausages.

4. Also justify the utilization of different salt in  sucuk preparation.

5. In the result portion, there is requirement to discuss with other researchers and also add the possible reasons so that readers can get possible benefits. This also helps to know about the chemistry behind utilizations of these salts.

6. Conclusion is just the summary of results. Please talk about the big picture and the major findings of the work. What's new that this paper offers to the readers?

 Author Response

Dear Editor,

We appreciate your sincere and helpful comments on this manuscript. The manuscript has been carefully rechecked, and changes have been made in accordance with the reviewers’ suggestions. The detailed corrections are listed below point by point and the changes are attached to the text in red color. 

Thank you very much for your attention to our manuscript.

Best regards,

Reviewer 1:

Article may be accepted after addressing the following comments:

  1. Add some numerical data in the abstract portion.

-Added.

  1. Add some keywords related to the properties studied.

-Added.

  1. Rewrite the introduction portion, start with Sucuk, then discuss the role of different salts on the sausages.

-Revised.

  1. Also justify the utilization of different salt in  sucuk preparation.

-Revised.

  1. In the result portion, there is requirement to discuss with other researchers and also add the possible reasons so that readers can get possible benefits. This also helps to know about the chemistry behind utilizations of these salts.

-Revised.

  1. Conclusion is just the summary of results. Please talk about the big picture and the major findings of the work. What's new that this paper offers to the readers?

-Revised.

Reviewer 2 Report

Dear Authors,

Thank you for submitting your manuscript titled "Partial replacement of NaCl by KCl, MgCl2, and CaCl2 chloride salts in the production of sucuk: effect on volatile compounds, lipid oxidation, microbiological, and sensory properties". The topic is pertinent, and the aim to understand the impact of substituting NaCl with various chloride salts on sucuk quality is commendable. After careful review, we have noted several areas that could benefit from further clarity and support. Below are our detailed comments and suggestions:

Introduction:

- Line 42: The terminology "may occur to unacceptable effects" is somewhat ambiguous. Could you specify or define what "unacceptable effects" entails in this context?    

- Line 50: The claim that sucuk "may pose a risk to many consumer groups" is broad. Could you elaborate on the specific consumer groups or conditions under which sucuk's sodium content might present risks?

Results and Discussion:

- Line 213-219: This section highlights the TBARS value and the non-impact of the salt mixture on it. Yet, a later reference to another study suggests a contrasting finding about CaCl2. It would enhance clarity if this reference was more seamlessly integrated with the findings from the current study or if any discrepancies were explained.

- Line 235-237: The mention of Kaban et al. [8] suggesting adverse sensory effects when replacing more than 50% of NaCl with KCl might need more context. If the conditions of the referenced study differ substantially from your own, this could mislead readers. Consider providing context or clarifying the relevance.

- Line 262-267: The discussion on hexanal and its prevalence among different sample groups concludes that different chloride salts did not impact hexanal, drawing parallels with TBARS values. If the connection between TBARS values and hexanal was not established earlier in your manuscript, this might confuse readers.

Author Response

Reviewer 2:

Dear Authors,

Thank you for submitting your manuscript titled "Partial replacement of NaCl by KCl, MgCl2, and CaCl2 chloride salts in the production of sucuk: effect on volatile compounds, lipid oxidation, microbiological, and sensory properties". The topic is pertinent, and the aim to understand the impact of substituting NaCl with various chloride salts on sucuk quality is commendable. After careful review, we have noted several areas that could benefit from further clarity and support. Below are our detailed comments and suggestions:

Thank you for your comments

Introduction:

- Line 42: The terminology "may occur to unacceptable effects" is somewhat ambiguous. Could you specify or define what "unacceptable effects" entails in this context?    

-Revised.

- Line 50: The claim that sucuk "may pose a risk to many consumer groups" is broad. Could you elaborate on the specific consumer groups or conditions under which sucuk's sodium content might present risks?

-Revised.

Results and Discussion:

- Line 213-219: This section highlights the TBARS value and the non-impact of the salt mixture on it. Yet, a later reference to another study suggests a contrasting finding about CaCl2. It would enhance clarity if this reference was more seamlessly integrated with the findings from the current study or if any discrepancies were explained.

-Revised.

- Line 235-237: The mention of Kaban et al. [8] suggesting adverse sensory effects when replacing more than 50% of NaCl with KCl might need more context. If the conditions of the referenced study differ substantially from your own, this could mislead readers. Consider providing context or clarifying the relevance.

-Revised.

- Line 262-267: The discussion on hexanal and its prevalence among different sample groups concludes that different chloride salts did not impact hexanal, drawing parallels with TBARS values. If the connection between TBARS values and hexanal was not established earlier in your manuscript, this might confuse readers.

-Removed and revised.

Round 2

Reviewer 1 Report

The author addresses all comments I suggested, and the article may be accepted.